# The Clinical Effects of Type 2 Diabetes Patient Management Using Digital Healthcare Technology: A Systematic Review and Meta-Analysis

**DOI:** 10.3390/healthcare10030522

**Published:** 2022-03-13

**Authors:** Ji-eun Kim, Tae-shin Park, Kwang Joon Kim

**Affiliations:** College of Pharmacy, Mokpo National University, Youngsan-ro 1666, Cheonggye-myeon, Mokpo 58554, Korea; kimjiji427@naver.com (J.-e.K.); ptsxotls@naver.com (T.-s.P.)

**Keywords:** digital healthcare technology, type 2 diabetes, HbA1c

## Abstract

The disease control rate is very low (at less than 30%) for diabetes. The use of digital healthcare technology is increasing recently for continuous management in daily life. In this study, a meta-analysis was conducted to evaluate the clinical effects of digital healthcare technology for patients with type 2 diabetes management. For a review of the literature, databases such as PubMed, Embase, and Cochrane Library were searched using Medical Subject Heading (MeSH) terms published up to 9 August 2021. As a result, 2354 articles were identified, and 12 randomized controlled trial articles were finally included. Digital healthcare technology combined management for type 2 diabetes significantly decreased HbA1c (*p* < 0.00001, standardized mean difference (SMD) = −0.49) and marginally decreased triglyceride, compared with usual care (*p* = 0.06, SMD = −0.18). However, it did not significantly affect BMI (*p* = 0.20, SMD = −0.47), total cholesterol (*p* = 0.13, SMD = −0.19), HLD-C (*p* = 0.89, SMD = −0.01), LDL-C (*p* = 0.95, SMD = −0.01), systolic BP (*p* = 0.83, SMD = 0.03), or diastolic BP (*p* = 0.23, SMD = 0.65), compared with usual care. These results indicate that digital healthcare technology can improve HbA1c and triglyceride levels of type 2 diabetes patients. Further well-designed randomized controlled clinical trials are needed to confirm the clinical effect of digital healthcare technology.

## 1. Introduction

Diabetes is among the top 10 global causes of mortality in adults, with four million deaths estimated globally in 2017 [1]. The global prevalence of diabetes was estimated at 9.3% in 2019 and was predicted to rise to 10.2% by 2030 [2]. According to “Diabetes Fact Sheet in Korea 2020” of the Korean Diabetes Association, the prevalence of diabetes mellitus in Korea was high in 2018 (13.8% in those over 30 years old and 27.6% in those over 65 years old) [3]. Recently, new medicines such as dipeptidyl peptidase-4 (DPP4) inhibitor, sodium-glucose co-transporter-2 (SGLT-2) inhibitor, and glucagon-like peptide-1 (GLP-1) agonist for type 2 diabetes mellitus (T2DM) are being introduced to the clinical field in addition to traditional medicines such as metformin, sulfonylurea, thiazolidinedione, and insulin. Among diabetic patients, 60.1% of those over 30 years old and 72.9% of those over 65 years old received pharmacotherapy in Korea [3]. However, the disease control rate for diabetic patients was very low, at less than 30% [3]. Specifically, less than 30% of adult diabetic patients had glycated hemoglobin (HbA1c) levels under 6.5%. Particularly, those with HbA1c levels over 6.5% but under 7% had continuously increased in 2016, 2018, and 2020 reports [3]. These data show that using pharmacotherapy alone is difficult to achieve sufficient effects in diabetes care. Thus, clinical practice guidelines of the American Diabetes Association and Korean Diabetes Association recommend starting lifestyle modification and monitoring pharmacotherapy response when patients are diagnosed with T2DM [4,5]. In other words, after patients are diagnosed with T2DM, proper modification of lifestyle including diet and physical activity is regarded as one of the most crucial factors along with the use of medicines. However, a recent Korean survey found that only 35.7% of diabetic patients regularly walked above 30 min per day [3]. This figure showed consistent decreases in 2016, 2018, and 2020 reports [3]. The ratio of obese patients (BMI ≥ 25 kg/m^2^) among diabetic patients had consistently increased, reaching up to 53.2% in 2020 [3]. Regarding diet associated with excess energy intake, it was found that protein and fat intake rates of diabetic patients were lower, but carbohydrate intake rate was higher than those of nondiabetic patients [3]. This indicates that continuous management of diet is necessary for diabetic patients. Based on these data of diabetic patients in Korea, inappropriate lifestyle habits can be one of the major reasons for the significantly lower disease control rates than treatment rates.

Although T2DM is a chronic disease that requires not just pharmacotherapy but also continuous lifestyle management, current diabetes management services mostly use laboratory test results and counseling data, which can be obtained restrictively at face-to-face meetings with medical specialists. However, such counseling data have limitations in accurately monitoring and providing intervention to modify lifestyles of patients who are not sincere in recording a diabetic diary. In fact, Kim et al. [6] have provided diabetes management service through regular face-to-face and telephone counseling to T2DM patients. They found that such a service had limitations in improving patients’ lifestyles. Along with the recent increase in smartphone owners, the development of 5G communication technology, and the development of Internet of Things (IoT) technology, digital healthcare technology using wearable and mobile devices is continuously developing, and continuous attempts are made to use such technology as a tool for caring patients with chronic diseases [7].

The objective of this systematic review and meta-analysis was to evaluate clinical effects of counseling and intervention service for diabetes management by healthcare providers (medical doctors, pharmacists, nurses, nutritionists, exercise therapists, etc.) using daily life data of T2DM patients collected through digital healthcare technology such as internet web, mobile phone apps, and connected devices. Based on the results of this study, we intend to find a more advanced management plan for type 2 diabetic patients.

## 2. Materials and Methods

This study was performed according to the Preferred Reporting Items for Systematic Reviews and Meta-Analyses (PRISMA) statement. Each process of this study was performed independently by two authors.

### 2.1. Search Method

PubMed, Embase, and Cochrane Library were selected as literature search databases to conduct a systematic literature review. The literature search was conducted for papers published until 9 August 2021. We used the PICO method to elaborate a specific key question suitable for the purpose of this study. Patient population (P): type 2 diabetes patients; Intervention (I): digital healthcare technology by healthcare providers; Comparison (C): usual patient care; Outcomes (O): HbA1c, BMI, LDL-C, HDL-C, blood pressure; Study design (SD): randomized controlled trials (RCTs).

We searched databases using Medical Subject Headings (MeSHs) and free-text terms combined with Boolean operators “AND” and “OR”, etc. (Appendix A).

### 2.2. Study Selection and Quality Assessment

Among studies searched from each database, only full-text articles designed as “Randomized clinical trial” and written in English were included. We screened and included studies to evaluate how healthcare providers applied to improve clinical outcomes for Type 2 diabetes patients. Duplicated studies between databases were excluded using EndNote 20 program. Studies unrelated to the purpose of this study were also excluded by screening titles and abstracts. Studies without HbA1c data, the primary outcome to be analyzed in this study, were also excluded by a full-text review. Two authors independently performed study selection and data extraction. A third author resolved any conflicts occurring through mutual consultation between authors and made final decisions. Two authors assessed the quality of each study and ultimately selected studies using the Cochrane’s risk of bias (RoB) tool [8]. RoB has seven domains: two selection bias, performance bias, detection bias, attrition bias, reporting bias, and other bias. Each domain was scored as “high risk”, “low risk”, or “unclear risk” according to the degree of the risk of bias. If it was difficult to identify the risk of bias, the study was assessed as having an “unclear risk of bias”. Publication bias of selected studies was assessed using a funnel plot.

### 2.3. Data Extraction

From each study, data of HBA1c as the primary outcome and body mass index (BMI), total cholesterol, triglyceride, low-density lipoprotein cholesterol (LDL-C), high-density lipoprotein cholesterol (HDL-C), systolic blood pressure, and diastolic blood pressure as secondary outcomes were extracted.

### 2.4. Data Synthesis and Statistical Analysis

Review manager 5.4 and R studio Version 1.4.1717 were utilized for data analysis. Since the extracted data in this study were continuous variables, standardized mean difference (SMD) was weighted by the number of study subjects of the intervention group and the control group in each study. Mean and standard deviations were calculated with 95% confidence intervals (CIs). Results are presented as a forest plot using the random effect model. Heterogeneity of results was assessed using Higgin’s I^2^: 0% ≤ I^2^ ≤ 40%, “may not be important”; 30% ≤ I^2^ ≤ 60%, “may represent moderate heterogeneity”; 50% ≤ I^2^ ≤ 90%, “may represent substantial heterogeneity”; 75% ≤ I^2^ ≤ 100%, “considerable heterogeneity” [9].

## 3. Results

### 3.1. Search Results

A total of 2354 studies were retrieved from PubMed, Embase, Cochrane Library in August 2021. After excluding non-RCT, non-trial, and duplicate studies, 323 studies remained. After secondarily excluding 99 studies not eligible for full-text criteria, the remaining 224 studies were screened for titles and abstracts. Finally, 12 studies were found to be eligible for analysis in this study (Figure 1) (Table 1).

### 3.2. Study Characteristics and Quality Assessment

The country, study design, study length, intervention patients, comparison patients, types of tools for intervention, contents of intervention, and clinical outcome measurements of the finally selected studies are summarized in Table 1. The meta-analysis was performed on a total of 1362 patients (digital healthcare: 686 patients, usual care: 676 patients) in the 12 studies. As a result of the quality assessment of the 12 studies, studies using a random number generated by a computer were assessed as having a “low risk” of selection bias. They were assessed as “unclear risk” if it was difficult to identify the appropriateness of a randomized method, or if the method was not described. All studies were assessed as “unclear risk” of performance bias because there was not enough evidence to evaluate the effect of a blind test. In cases in which there was no dropout during the intervention period, or it was determined that the missing value would not significantly affect the effect size, studies were assessed as having a “low risk” of attrition bias. If outcomes presented in study protocols were excluded from study results, such studies were assessed as having a “high risk” of reporting bias (Figure 2). A funnel plot was expressed for the publication bias of selected studies (Figure 3).

### 3.3. Primary Outcome Analysis

As a result of a meta-analysis of the 12 studies to determine the reduction in HbA1c in the intervention group using digital healthcare technology, the intervention group showed a statistically significant reduction in HbA1c, compared with the comparison group (SMD: −0.49 [95% CI: −0.64, −0.33], I^2^ = 48%, *p* < 0.00001) (Figure 4).

### 3.4. Secondary Outcome Analysis

As a result of a meta-analysis of five studies presenting BMI levels to determine the effects of interventions on BMI, the intervention group did not show a statistically significant difference in BMI, compared with the comparison group (SMD: −0.47 [95% CI: −1.20, 0.25], I^2^ =95%, *p* = 0.20) (Figure 5). Results of a meta-analysis of three studies presenting total cholesterol levels showed that total cholesterol levels in the intervention group were not significantly different from those in the comparison group (SMD: −0.19 [95% CI: −0.43, 0.05], I^2^ = 41%, *p* = 0.13) (Figure 6). Results of a meta-analysis of three studies presenting triglyceride levels showed a marginally significant reduction in the intervention group, compared with the comparison group (SMD: −0.18 [95% CI: −0.37, 0.01], I^2^ = 0%, *p* = 0.06) (Figure 7). Results of a meta-analysis of three studies presenting LDL-C levels showed that LDC-L levels in the intervention group were not significantly different from those in the comparison group (SMD: −0.01 [95% CI: −0.30, 0.29], I^2^ = 52%, *p* = 0.95) (Figure 8). Results of a meta-analysis of three studies presenting HDL-C levels showed no statistically significant difference between the intervention group and the comparison group (SMD: −0.01 [95% CI: −0.21, 0.19], I^2^ = 0%, *p* = 0.89) (Figure 9). Results of a meta-analysis of five studies presenting systolic blood pressure levels showed no significant difference between the intervention group and the comparison group (SMD: 0.03 [95% CI: −0.26, 0.32], I^2^ = 69%, *p* = 0.83) (Figure 10). Results of a meta-analysis of five studies presenting diastolic blood pressure levels showed an increase in the intervention group, compared with the comparison group, although such increase was not statistically significant (SMD: 0.65 [95% CI: −0.41, 1.71], I^2^ = 97%, *p* = 0.23) (Figure 11).

## 4. Discussion

T2DM is a chronic disease that requires continuous management of lifestyle with pharmacotherapy. However, current diabetes management services for poorly controlled T2DM patients have limitations to modify diabetes management lifestyle. Along with the development of IoT and 5G communication technology, attempts are continuously being made to use digital healthcare technology as a tool for chronic diseases care, with notable strength [7,22,23]. Digital healthcare technology is usually used to help patients enhance and sustain their healthy behaviors such as physical activity, medication adherence, nutrition intake, and stress management. For healthcare providers, it can increase opportunities to contact patients, thus enabling closer monitoring.

In this study, we conducted a systematic review and meta-analysis to evaluate the clinical effects of counseling and intervention services by healthcare providers using patient-generated health records of T2DM patients collected through Internet websites, mobile phone apps, and connected devices.

As a result, the meta-analysis was performed on a total of 1362 patients in 12 studies. We found that digital healthcare technology for type 2 diabetes patient management did significantly decrease HbA1c, the primary outcome, compared with usual care (*p* < 0.00001, SMD = −0.49). The control group also showed a decrease in HbA1c. However, the decrease in the study group was larger than that in the control group. This finding indicates that digital healthcare technology is effective in improving clinical outcomes of T2DM patients. Similarly, Park et al. [24] systemically reviewed digital health interventions using telephones, web tools, and mobile apps by clinical pharmacists. Their recent study results found that mobile-based and web-based interventions improved clinical effects on lab values. On the other hand, all other secondary outcomes showed no significant results. As demonstrated in a previous study, these results might be because target patients were recruited based on HbA1c levels with deviations for other secondary outcomes [10].

For behavioral parameters, some studies evaluated self-efficacy, anxiety and depression, quality of life (QOL), diabetes knowledge, etc. [10,11,15,16,18,19,20]. QOL and self-efficacy scores were generally increased, although such increases in some of these results were not statistically significant. For example, one research in India with a low penetration rate (20%) of smartphones showed that the effectiveness of digital healthcare technology was insufficient due to the relatively low ability to use smartphones [14]. The use of mobile- and web-based devices can be considered a barrier for elderly patients or low-educated participants [25]. Therefore, education on the use of basic equipment such as smartphone applications and connected devices will be required continuously for digital healthcare technology to achieve more than a certain level of effect.

The results from this systematic review and meta-analysis of the clinical impact of digital healthcare technology intervention by healthcare providers demonstrated that closer and continuous monitoring by healthcare providers using digital healthcare technology, i.e., mobile-app-based and web-based interventions, may potentially help solve type 2 diabetes management challenges. Nevertheless, there are several limitations to this study. First, there may be selection bias because non-English publications were excluded. Second, there is a limitation in the generalization of the meta-analysis results of secondary outcomes because target patients were recruited based on HbA1c levels with deviations for other secondary outcomes. Third, recent study results published after the literature search were not reflected. Fourth, each study had different types of tools and contents for intervention, and individual ability in adopting this digital technology was not reflected. Fifth, this study did not assess the cost-effectiveness of digital healthcare technology intervention by healthcare providers. Therefore, the cost-effectiveness of digital healthcare technology intervention needs to be evaluated in future studies. Additionally, since more advanced devices such as wearable devices capable of continuous blood glucose measurement (CBGM) are being developed recently, it seems necessary to evaluate the study results using these new devices in the future.

## 5. Conclusions

Management for type 2 diabetes patients using digital healthcare technology significantly decreased HbA1c levels, compared with usual care (*p* < 0.00001, SMD = −0.49). It also marginally decreased triglyceride levels, compared with usual care (*p* = 0.06, SMD = −0.18). However, it did not significantly affect BMI, total cholesterol, HDL-C, LDL-C, systolic BP, or diastolic BP, compared with usual care. These results show that digital healthcare technology can decrease HbA1c and triglyceride levels of type 2 diabetes patients with improved clinical effects. However, further well-designed randomized controlled clinical trials are needed to prove and confirm the clinical effects of digital healthcare technology on T2DM patient management.

## Figures and Tables

**Figure 1 healthcare-10-00522-f001:**
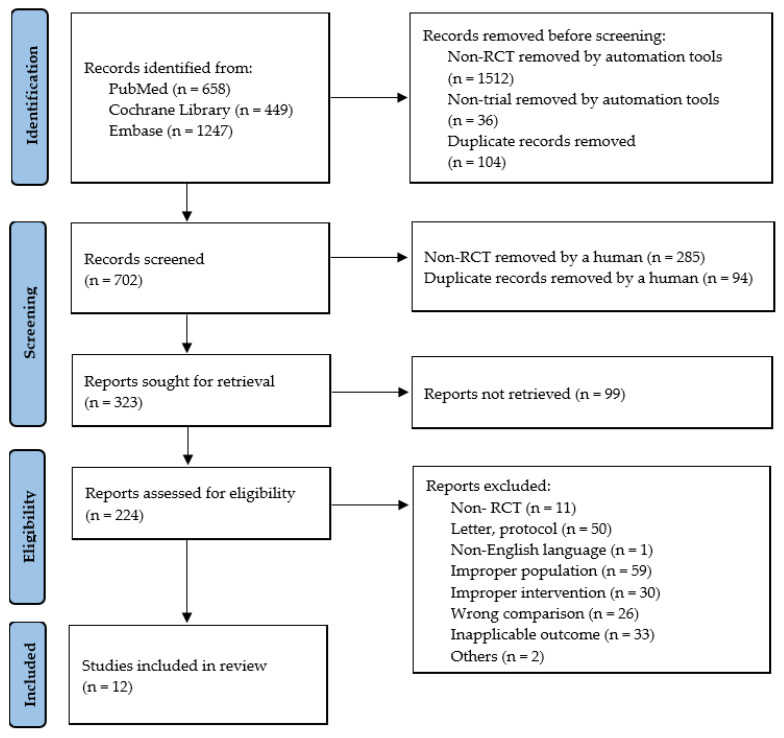
Flow diagram of literature search.

**Figure 2 healthcare-10-00522-f002:**
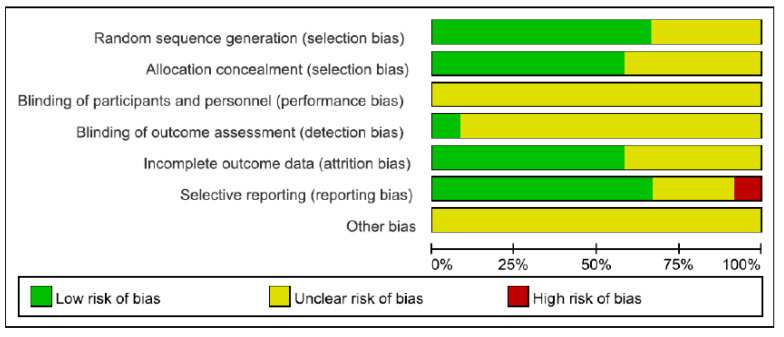
Risk of bias graph.

**Figure 3 healthcare-10-00522-f003:**
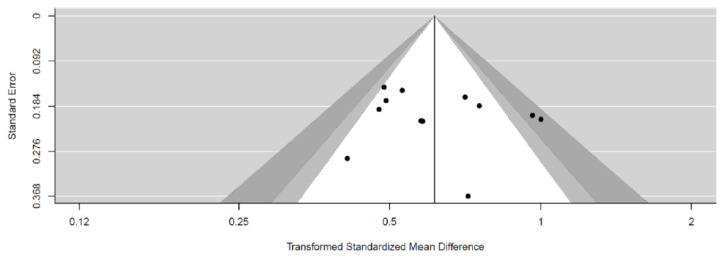
Check for funnel plot.

**Figure 4 healthcare-10-00522-f004:**
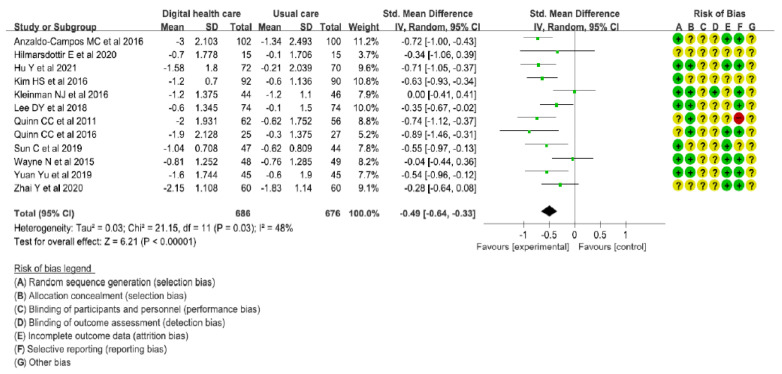
Forest plot for meta-analysis results of HbA1c [10,11,12,13,14,15,16,17,18,19,20,21].

**Figure 5 healthcare-10-00522-f005:**
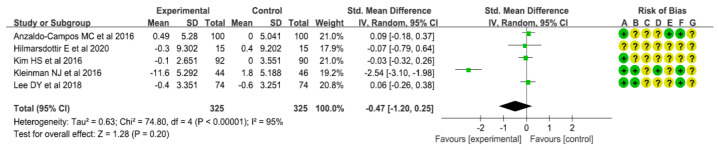
Forest plot for meta-analysis results on BMI [10,11,13,14,15].

**Figure 6 healthcare-10-00522-f006:**
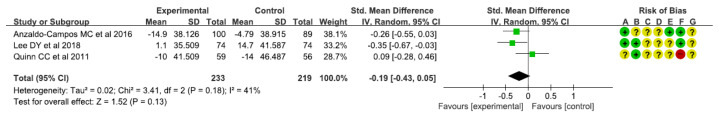
Forest plot for meta-analysis results on total cholesterol [10,15,16].

**Figure 7 healthcare-10-00522-f007:**
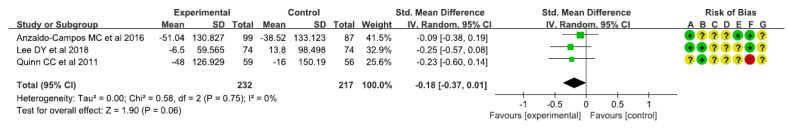
Forest plot for meta-analysis results on triglyceride [10,15,16].

**Figure 8 healthcare-10-00522-f008:**
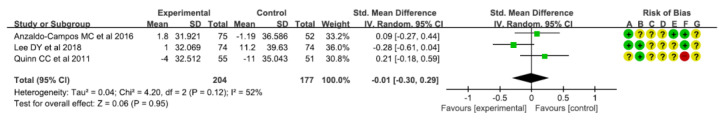
Forest plot for meta-analysis results on LDL-C [10,15,16].

**Figure 9 healthcare-10-00522-f009:**
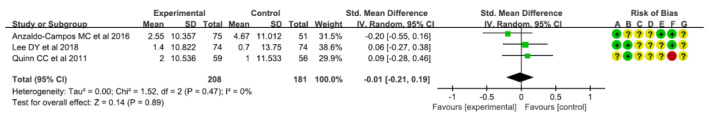
Forest plot for meta-analysis results on HDL-C [10,15,16].

**Figure 10 healthcare-10-00522-f010:**
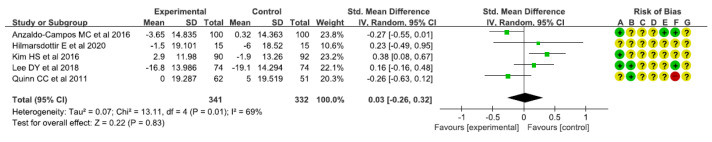
Forest plot for meta-analysis results on systolic blood pressure [10,11,13,15,16].

**Figure 11 healthcare-10-00522-f011:**
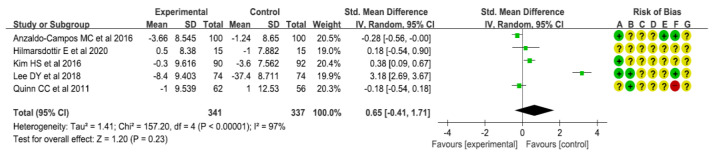
Forest plot for meta-analysis results on diastolic blood pressure [10,11,13,15,16].

**Table 1 healthcare-10-00522-t001:** Baseline characteristics of included studies evaluating digital healthcare technology interventions by healthcare providers.

Author	Location and Duration	Intervention; Digital Healthcare (*n*, Mean Age)	Comparison; Usual Care (*n*, Mean Age)	Healthcare Providers for Intervention	Type of Tools for Intervention	Contents of Intervention	Clinical Outcome Measurements
Anzaldo-Campos, M.C. et al. 2016 [10]	Mexico, Hosptial based, 10 months	*n* = 102, 51.5	*n* = 100, 52.5	(1) Physician (2) Nurse	Glucose meter with USB connection	(1) Tracking glucose level (2) Interactive surveys and text messages through the app (3) Educational brochures and videos through the app	HbA1C (%), total cholesterol, LDL-C ^1^ HDL-C ^2^, triglycerides, blood pressure (systolic, diastolic), BMI ^3^
Hilmarsdottir E. et al. 2020 [11]	Iceland, Hospital-based endocrine clinic, 6 months	*n* = 15, 50.9	*n* = 15, 51.5	Doctor	Smartphone application	(1) Guidance for a healthy lifestyle through the app (2) Individualized encouragement through the app	HbA1c (%), total cholesterol, triglycerides, HDL-C, LDL-C, weight, BMI, waist circumference, blood pressure (systolic, diastolic)
Hu, Y. et al. 2021 [12]	China, Hospital-based endocrine clinic, 6 months	*n* = 72, 50.04	*n* = 70, 52.21	(1) Endocrinologist (2) Nurse	Blood-glucose management platform	(1) Providing diabetes education (self-monitoring of blood glucose levels, dietary habits, medication timing, and physical activity) (2) Contacting patients through telephone or other online connections, if necessary	HbA1c (%), hypoglycemic events, ^4^ UACR, carotid plaque
Kim, H.S. et al. 2016 [13]	Korea, Hosptial based, 6 months	*n* = 92, 52.5	*n* = 90, 55.6	(1) Doctor (2) Nurse	Blood sugar monitoring through the Internet	(1) Tracking blood glucose levels and health conditions regularly (2) Recommendations on blood glucose control	HbA1C (%), FBG, FBG, BMI, LDL-C, HDL-C, total cholesterol, triglycerides, weight, blood pressure (systolic, diastolic)
Kleinman, N.J. et al. 2016 [14]	India, Hospital-based, 6 months	*n* = 44, 48.8	*n* = 46, 48.0	(1) Doctor (2) Health coach	Smartphone application, (m-Health ^5^ diabetes management platform)	(1) Reminding participants to complete missions every day (2) Automated follow-up to abnormal blood glucose tests (3) Regular responding to patient questions and system-generated alerts	HbA1C (%), FBG, BMI
Lee, D.Y. et al. 2018 [15]	Korea, Hospital-based, 6 months	*n* = 74, 51.4	*n* = 74, 52.6	(1) Endocrinologist (2) Nurse (3) Dietitian	Mobile application	(1) Tailored mobile coaching (2) Regular mobile messages (3) Communication through the app	HbA1c (%), BMI, blood pressure (systolic, diastolic), total cholesterol, triglycerides, HDL-C, LDL-C
Quinn, C.C. et al. 2011 [16]	USA, University Hospital-based, 12 months	*n* = 62, 52	*n* = 56, 53.2	Doctor	Mobile diabetes management software application and a web portal	(1) Receiving automated and real-time messages specific to the entered data (educational, behavioral, and motivational message) (2) Analyzing patient data based on standards of care	HbA1C (%), blood pressure (Systolic, Diastolic), LDL-C, HDL-C, triglycerides, total cholesterol
Quinn, C.C. et al. 2016 [17]	USA, University Hospital-based, 12 months	*n* = 25, 59.0	*n* = 27, 59.5	Physician	Mobile diabetes management software application	(1) Receiving automated and real-time messages specific to the entered data (educational, behavioral, and motivational message) (2) Intermittently reviewed by virtual case managers	HbA1C (%)
Sun, C. et al. 2019 [18]	China, University Hospital-based, 6 months	*n* = 44, 67.9	*n* = 47, 68.04	(1) Medical team (2) Dietitian	mHealth management system based on mobile phone	(1) Sending medical advice and reminders to patients (2) Guidance for blood glucose monitoring and dietary advice based on the individual blood glucose levels (3) Guidance related to aerobic and resistance-based exercise	HbA1c (%), FBG, total cholesterol, triglycerides, HDL-C, LDL-C, BMI, blood pressure (systolic, diastolic)
Wayne, N. et al. 2015 [19]	Canada, Primary care Health-center-based, 6 months	*n* = 48, 53.1	*n* = 49, 53.3	Health coach (behavior-change counseling specialist with expertise in chronic disease management)	Smartphone application	(1) Tracking key metrics (blood glucose levels, exercise frequency, exercise duration, exercise intensity, food intake, and mood) (2) Communicating with a health coach at any time (3) Communicating with a health coach at scheduled phone contact and during in-person meetings	HbA1C (%), weight, BMI, waist circumference
Yu, Y. et al. 2019 [20]	China, University Hospital-based endocrine clinic, 6 months	*n* = 45, 50.3	*n* = 45, 51.4	Physician	Smartphone application	(1) Virtual education through the app (diet library, video and picture demonstration for exercise, information about blood glucose monitoring, and latest guidelines) (2) Automatically generated message to the patient and notification to clinicians if the blood glucose value was found abnormal value (3) Answering patient’s questions and offering recommendations based on individual data through the app	HbA1c (%), FBG, 1.5-anhydroglucitol, proportions of patients achieving HbA1c < 7.0%
Zhai, Y. et al. 2020 [21]	China, Hospital-based, 6 months	*n* = 60, 54.12	*n* = 60, 55.64	(1) Physician (2) Nures	Smartphone application	(1) Providing support for diabetes self-management (diet advice, emotional management, and medication guidance) (2) Reviewing blood glucose data (3) Providing online instruction (diet, exercise, blood glucose monitoring, insulin injection) and answering patient’s questions (4) Analyzing the causative factors of abnormal blood glucose and giving advice on how to avoid them	HbA1c (%)

^1^ Low-density lipoprotein, ^2^ high-density lipoprotein, ^3^ body mass index, ^4^ UACR, urine albumin-to-creatinine ratio, ^5^ mobile health, fasting blood glucose.

## Data Availability

Data are contained within the article.

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
