# Peer review of "The Clinical Effects of Type 2 Diabetes Patient Management Using Digital Healthcare Technology: A Systematic Review and Meta-Analysis"

_healthcare, 2022, doi:10.3390/healthcare10030522_

Round 1

Reviewer 1 Report

Dear authors,

This manuscript aims to analyze the effect that technologies have on chronic patients with type 2 diabetes mellitus. The follow-up of these patients is very important for good control of the disease. Therefore the manuscript is pertinent.

The manuscript is interesting but I need you to answer some questions:

INTRODUCTION

The authors should compare the epidemiological data of Korea with other countries.

MATERIALS AND METHODS

Search Method:

  • Search is out of date. Authors should review if there are new publications that meet the inclusion criteria.
  • Authors must specify the search equation and Boolean operators used. The search must be reproducible and if you don't see how the operators and descriptors are combined, it can't be done.
  • Authors should perform a reverse search to broaden the results.

Inclusion & exclusion criteria:

  • In the inclusion / exclusion criteria there is information that the authors have not specified:
  • What languages were analyzed?
  • How long does the search go back?

DISCUSSION

References of the results cannot be used in the discussion. This is incorrect. Authors should remove these references.

The authors have not explained how to apply the research to clinical practice.

The authors have not included study limitations.

Reviewer 2 Report

  • Very well conducted analysis and written article
  • Note: Medical Subject Headings is abbreviated MeSH
  • Please share your search string for further reproducibility of your approach
  • Reference the RoB tool if possible (in methods)
  • Sentence edit: line 197-198 (second sentence in discussion)

Reviewer 3 Report

A very interesting study that employed rigorous methods for conducting a review, e.g., systematic approach using PRISMA protocol and meta-analysis. Some of the English is a bit awkward and there are some typos (Line 68, objective is misspelled).

The abstract (lines 12-13) states the goal of the study as “to evaluate clinical effects of digital healthcare technology for patients with type 2 diabetes management.” In contrast, the text (lines 67-71) explains the goal of the study “to evaluate clinical effects of counseling and intervention service by healthcare providers (medical doctors, pharmacists, nurses, nutritionists, exercise therapists, etc.) using daily life data of T2DM patients collected through digital healthcare technology such as wearable devices and mobile phone apps.” The paper addresses the goal provided in the abstract but not the goal provided in the text. The reason is that the “counseling and intervention services” provided are not discussed.

Table 1: Add interventions used in each study (medication, dietary modification, physical activity recommendation, electronic monitoring, etc.). Also, suggest adding which type of healthcare provider(s) were involved in the delivery of the interventions.

Line 68: “The objecitve of this systematic review and meta-analysis was to evaluate clinical effects of counseling and intervention service by healthcare providers…” It is unclear to me what you mean by “intervention service.”  Suggest more precise language; for example, does it mean lifestyle changes such as diet and activity, or does it mean pharmaceutical interventions?

Lines 71-72: “Based on results of this study, we intend to find a more advanced management plan for type 2 diabetic patients.”  This intent is not addressed in the discussion or the conclusion. Did the results yield a more advanced plan?

In the discussion, add information about the various wearable devices and mobile apps such as studies evaluating their general efficacy. Also, need to consider pros and cons of different devices for tracking this population. You touch on it when you note the study from  India and lack of knowledge of smart phone usage; however, what you’ve included needs to be expanded on.

To achieve your goal of evaluating the clinical effects of counseling and intervention service by healthcare providers, you also need to discuss which specific interventions appear to be best suited for tracking via wearables of smart phone apps.

Lines 229-232: This paragraph seems out of context. Given that no information was provided on the specific intervention components or healthcare team members, you have not provided evidence to support this statement. I suggest deleting these lines.

Lines 228-240: This conclusion goes too far given the specifics of the intervention components were not evaluated in the study.

Reviewer 4 Report

The authors present an interesting meta-analysis on the clinical effectiveness of healthcare digital technologies in people with type II diabetes mellitus. The paper is well written and follows the recommendations of the PRISMA statement. However, some aspects deserve to be revised by the authors.
Below, I list my suggestions for the authors hoping that they will help them improve their manuscript.

1. Introduction. In this section, the authors only present data on diabetes and the use of technological devices from Korea. This scope would make sense if the study was framed exclusively in Korea. Since this is a meta-analysis, I suggest the authors incorporate global data.

2. Materials and Methods.

2.1. First, outside of any subsection (2.X.), indicate the study design and the PRISMA statement has been followed.

2.2. Try to adjust the manuscript sections to those recommended by the PRISMA statement. For example, this statement suggests starting with the eligibility criteria and information sources.

2.3. Specify in the text or supplementary material the search strategies used in each database.

2.4. Explicitly formulate the PICO question.

2.5. Did the authors use any tool for automatic selection or screening of records (e.g., a reference manager)? If so, this should be indicated in the text.

2.6. Inclusion criteria should be described in more detail. In the current version of the manuscript, the authors only refer to the type of study design (randomized clinical trial) and the language of publication (English). Other relevant criteria referring to the technological intervention type, participant profile (age, pathology, etc.), publication dates, etc., are missing.

2.7. Did the authors include the term "HbA1c" as a descriptor in their searches?

2.8. Incorporate the corresponding reference for the "Cochrane's risk of bias (RoB) tool" (lines 93-94).

2.9. Were studies performed in populations with type I diabetes mellitus excluded?

2.10. The authors should analyze the possible risk of publication bias, for example, using Egger's regression test.

3. Results

3.1. Page 3, lines 180-120. Screening by title and abstract is one of the first manual screens to be performed and is followed by a full-text review of those records that pass the screening. This aspect is not clear in the text.

3.2. Table 1

3.2.1. Some data corresponding to the number of patients per study does not coincide with that shown in Figure 4 (are lower). Review data from the following studies: Anzaldo-Campos MC et al. 2016 (102/100 instead of 89/92); Lee DY et al. 2018 (74/74 instead of 72/64); Quinn CC et al. 2011 (62/56 instead of 56/51); Sun C et al. 2019 (47/44 instead of 44/47); Zhai Y et al. 2020 (60/60 instead of 60/58). The sum of the n in the table results in 1,316 patients, whereas the text reports 1,362. The difference is due to the dance of the figures indicated.

3.2.2. I suggest eliminating the column "Study design" since all the studies are randomized clinical trials. If the authors wish, they can include this information in the table title.

3.2.3. In the column "Types of tools for intervention", specify whether data recording is automatic (e.g., activity bracelet) or manual by the user.

3.3. Lines 130-131. Of the 1,362 patients included in the meta-analysis, specify how many belonged to the experimental condition and how many to the control. Indicate the mean age of the participants.

3.4. In the text, I miss a more detailed description of the intervention types included in the meta-analysis.

3.5. Unify the terminology used in the forest plots. " Digital health care vs. Usual care" or "Experimental vs. control".

4. Discussion. Describe the study limitations.

5. Was the meta-analysis registered in PROSPERO or any equivalent registry system?

Round 2

Reviewer 3 Report

Table 1: Needs editing, so awkward English. For Contents of Intervention need to be paraphrased in your own words. I suggest using bullets rather than full sentences, so that you avoid plagiarism. Also, Anzaldo-Campos and Kim, Contents of Intervention—there is no number “1)”

Risk of bias:  Please double check blinding for participants and researchers. Based on the type of study, I would imagine it was not possible to implement blinding. For example, participants were using the devise. If there was not a control group who was also using the devise, then the participants were not blinded to the protocol.

Line 245-246: “has a positive potential for solving the type 2 diabetes management problems” Suggest revising to: may potentially help solve type 2 diabetes management challenges.
